# LPD-YOLOv7-tiny: An enhanced lightweight YOLOv7-tiny model for real-time potato quality detection

Hong Yu[1], Jiaxuan Hao[2], Yongbo Li[3]*

1 College of Agricultural Engineering,Jiangsu Agri-animal Husbandry Vocational College, Taizhou, China, 2 College of Art and Design, Nanning Normal University, Nanning, China, 3 College of Intelligent Manufacturing, TaiZhou Polytechnic college, Taizhou, China

* yzdxlyb@163.com

## Abstract

To solve the problems of low detection accuracy, large model size and slow reasoning speed of existing potato quality detection models, this paper proposes LPD-YOLOv7-Tiny, a lightweight potato sprout and spoilage detection model based on YOLOv7-Tiny. The proposed model introduces MobileNetV3 small, BiFormer, SimAM, and the Focal-EIOU loss function. MobileNetV3 small greatly reduces the number of parameters and computational complexity of the model, BiFormer enhances the multi-scale feature fusion capability of the model, and the SimAM module effectively suppresses irrelevant information and strengthens local features. The Focal-EIOU loss function improves the model's attention to difficult classification samples and enhances its bounding box regression capability. LPD-YOLOv7-Tiny achieves excellent detection performance on potatoes under complex background conditions: mAP is increased to 90.3%, the number of parameters is reduced to 5.8 MB, the number of computations is reduced to 10.1 G, and the inference speed is increased to 142.5 fps. Compared with mainstream detection models such as the YOLO Basic series, SSD and speed-RCNN, LPD-YOLOv7-Tiny achieves significantly improved performance in terms of detection accuracy, positioning capability and computational efficiency, indicating it has wide application potential in resource-constrained and high-precision scenarios.

## Introduction

In the context of the potato becoming a staple food, potato products have developed rapidly with a continuously extended industrial chain. In particular, vigorous development momentum has occurred in processing linkages. However, certain deficiencies remain with respect to the inspection and testing capabilities of the corresponding products [1–3]. The traditional detection of surface defects on potatoes relies mainly on manual picking, which has the disadvantages of low efficiency, strong subjectivity

**Data availability statement:** All relevant data are within the manuscript and its Supporting Information files. All relevant data are on Figshare: https://doi.org/10.6084/m9.figshare.29458949.

**Funding:** This study was supported by the Jiangsu Education Department, the Taizhou Association for Science and Technology, and Jiangsu Agri-animal Husbandry Vocational College in the form of project funds to Li Y and Yu H (24KJB510045; NSF2023ZR12). The specific roles of this author are articulated in the 'author contributions' section. The funders had no role in study design, data collection and analysis, decision to publish, or preparation of the manuscript.

**Competing interests:** The authors have declared that no competing interests exist.

and high cost and is difficult to adapt to industrial development [4–6]. How to achieve rapid and non-destructive testing has gradually become a key demand and research hotspot in potato quality control.

With recent advancements in technology, deep learning methods have been increasingly applied to defect detection for agricultural products and achieved remarkable progress [7]. Classic object detection algorithms such as Fast R-CNN and the YOLO series have been widely applied to defect detection for agricultural products [8–11]. In Fast R-CNN [12], a two-stage object detection model, candidate regions are first generated, and then samples are classified and regressed through convolutional neural networks. Wang et al. [13] used transfer learning to train the Faster R-CNN model and achieved a 98.7% accuracy rate on potato surface defect detection. Wei et al. [14] adopted the ResNet50 feature network to replace VGG16 in their original framework, solving the problems of easy false and missed small target detection to improve the efficiency and accuracy of cherry surface defect detection. Zhang et al. [15] developed an apple defect region detection algorithm based on Fast R-CNN. They replaced the feature extraction network from VGG16 with ResNet50 and added three suggestion boxes of different scales, which quickly and accurately identified apple defect regions under complex background conditions. Liu et al. [16] improved the feature extraction network in Faster R-CNN by fusing the feature extraction network of ResNet50 with FPN, which enhanced the detection ability of small targets such as potato sprouting and damage. Although the two-stage detection algorithm achieved high accuracy, it lacked processing speed and real-time performance.

YOLO [17] is a single-stage detection method that extracts features directly from a network for classification and target location. YOLO not only improves training speed but also reduces model complexity. Tian et al. [18] developed the improved YOLOv3 algorithm specifically for detecting the growth stages of apples. YOLOv3 achieved high efficiency in different detection scenarios, and the average processing time of each image does not exceed 0.31 seconds. Li et al. [19] designed a lightweight potato surface defect detection network based on YOLOv5s. By integrating three modules: the Coordinate Attention Mechanism (CA), Adaptive Spatial Feature Fusion (ASFF), and Atrous Spatial Pyramid Pooling (ASPP), the model's reasoning ability in cross-scale feature fusion was enhanced. Zhang et al. [20] integrated the CBAM attention mechanism and the BiFPN feature fusion structure, achieving dual enhancement of the YOLOv5s framework and significantly improving the performance of potato bud eye detection. Wu et al. [21] proposed the Flaw-YOLOv5s model, which replaced the original Conv in the YOLOv5s network with depth-wise separable convolution (DWConv) to reduce computational cost and the number of parameters, and replaced SPPF in the backbone with SPPELAN to optimize the detection of small potato targets. Liu et al. [22] optimized the YOLOX model by introducing the Focal loss function and the CBAM attention mechanism, significantly improving the performance of cherry defect detection and grading, achieving an average defect detection accuracy rate of 97.59%. Feng et al. [23] proposed an enhanced deep learning model based on YOLOX. By introducing the residual connection of the

neck network, cascading, an attention mechanism module and an optimized loss function, real-time multi-type detection of orange surface defects was achieved. Lu et al. [24] developed the YOLO-FD model, which combined the PSO-ELM algorithm and a dual-camera data acquisition system to achieve effective detection of citrus peel defects and accurate evaluation of fruit morphology. Yao et al. [25] proposed a kiwifruit defect detection model based on the YOLOv5 algorithm. By adding a small target detection layer, embedding the SE layer to enhance feature extraction, introducing the CIoU loss function to improve regression accuracy, and using the cosine annealing algorithm to optimize the training process, the performance of the model was improved.

In conclusion, scholars have conducted extensive research on the identification of surface defects in agricultural products and achieved good results. However, due to the wide variety and diverse forms of surface defects on potatoes and the common problems of complex structure and large numbers of calculations in existing detection models, both recognition performance and detection accuracy have decreased significantly for target detection in complex environments [26–30]. To overcome the limitations of these technologies, this study uses YOLOv7-tiny to construct a lightweight detection algorithm with high precision and low computational load through model structure improvement and feature fusion optimization. The proposed model's enhanced application performance on the detection of potato sprouting and spoilage meets the actual demands of the potato industry for rapid, accurate and efficient detection.

## Introduction to the YOLOv7-tiny algorithm

The network architecture of YOLOv7-Tiny shown in Fig 1 adopts a modular design consisting of three main parts: the feature extraction network (backbone), the feature fusion network (neck), and the feature detection output layer (head). In the feature extraction network (backbone) section, YOLOv7-Tiny adopts multi-level ELAN-Tiny structural units. Each ELAN-Tiny unit is composed of multiple groups of CBLs (Conv-BN-Leakyrelus) in series and feature concatenation (Concat). Combined with multiple MaxPooling operations, the receptive field of the feature map is effectively expanded. This efficient aggregation unit optimizes the gradient flow path, enhances the expressive ability of deep semantic features, and promotes stable convergence during the network training process. In the feature fusion network (neck) section, the network introduces a lightweight spatial pyramid pooling fusion module (SPPCSPC-Tiny), which integrates global and local information through multi-scale MaxPooling branches, enhancing the detection robustness of targets at different scales. The neck section simultaneously contains multiple groups of ELAN-Tiny units and CBL modules. Through upsampling and multiple Concat operations, the feature information of the shallow and deep layers is fully integrated, improving detection accuracy for small targets and complex scenarios. In the feature detection output layer (head) section, YOLOv7-Tiny performs convolution transformation on the fused multi-scale feature maps through multiple sets of CBL modules. Adjust the channel dimension and output the final detection results, including the category probabilities and bounding box regression values at each scale.

## Proposed method

### The MobileNetV3 lightweight backbone of the small network

To further reduce the number of parameters and computational complexity of the YOLOv7-tiny model, its backbone network is replaced by the MobileNetV3 small network. A lightweight neural network, MobileNetV3 [31] is divided into two versions, small and large; the small version is more lightweight and especially suitable for environments with limited computational resources. MobileNetV3 small is based on Neural Architecture Search (NAS), which combines deep separable convolution and linear bottleneck inverse residual structure to significantly improve detection efficiency. Meanwhile, the network introduces the SE attention mechanism, which can dynamically adjust the weights of different channels so the network pays more attention to information containing key features. In addition, optimizes the Swish activation function, effectively improving the accuracy and efficiency of target detection, especially for real-time detection tasks in mobile devices and embedded systems.

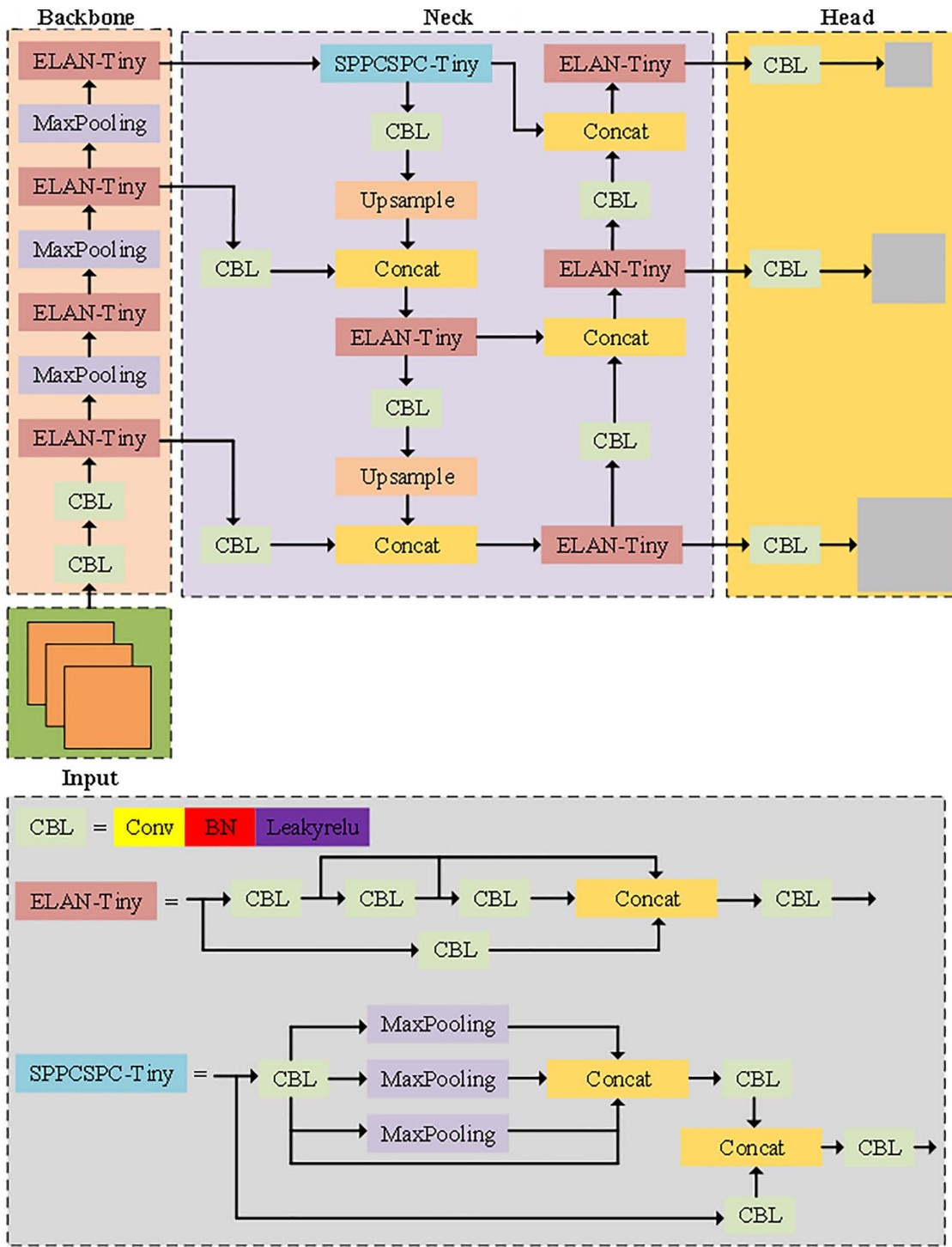

**Fig 1. YOLOv7-tiny model structure diagram.** Note: This network consists of three parts: the Backbone, the Neck and the Head. In the figure, different colored boxes represent different modules, and arrows indicate the direction of feature flow. The lower half of the figure shows the submodule structure diagrams of CBL, ELAN-Tiny and SPPCSPC-Tiny. The modular design facilitates the lightweight deployment of the model and enhances its real-time detection performance in resource-constrained environments.

The core component of MobileNetV3 small is the Bneck module shown in Fig 2. The Bneck module consists of depth-separable convolution, the SE attention mechanism, and shortcut connections. The input feature map (C × H × W) is first upscaled by a 1 × 1 convolutional layer to extract more feature dimensions. Then, the feature map is convolved with a 3 × 3 depth-separable convolution, keeping the number of channels constant, thus reducing computational complexity while retaining key information. The feature map is subsequently pooled by global averaging to generate a tensor of size C × 1 × 1 representing the global weight of each channel, which is multiplied with the corresponding input features to adaptively enhance key information. Finally, a 1 × 1 convolutional layer is used for dimensionality reduction to extract deeper feature information, and shallow features are passed to deeper layers through shortcut connections, thus realizing feature reuse and improving detection efficiency and accuracy.

## Incorporating the BiFormer attention mechanism

To improve the detection performance of the model for occluded targets, the BiFormer [32] module is introduced into the feature fusion layer (neck) of YOLOv7-Tiny. BiFormer is a variant of Transformer that realizes flexible content-aware computation with dynamic query-aware sparsity through two-layer routing. Its dynamic attention mechanism consists of three steps, as shown in Fig 3. First, the input feature graph is divided into multiple regions, and the query (Q), key (K) and value (V) tensors are obtained by linear mapping. Next, a directed graph is constructed using an adjacency matrix to determine the key-value pair relationships of different regions and capture local and global information. Finally, fine-grained feature focusing is performed through the routing index matrix ($I^r$), and sparse sampling skips the regions that are irrelevant to the target, which reduces the computational burden while retaining important detail information. By filtering out irrelevant regions, this mechanism achieves a balance between the retention of detailed features and computational efficiency and effectively reduces leakage and false detection.

In terms of multi-scale feature fusion, BiFormer further improves the accuracy of target detection by realizing attention to different scale targets in the feature map through dynamic routing and selecting the most relevant regions in a sparse manner. The process is shown in Formulas (1–2):

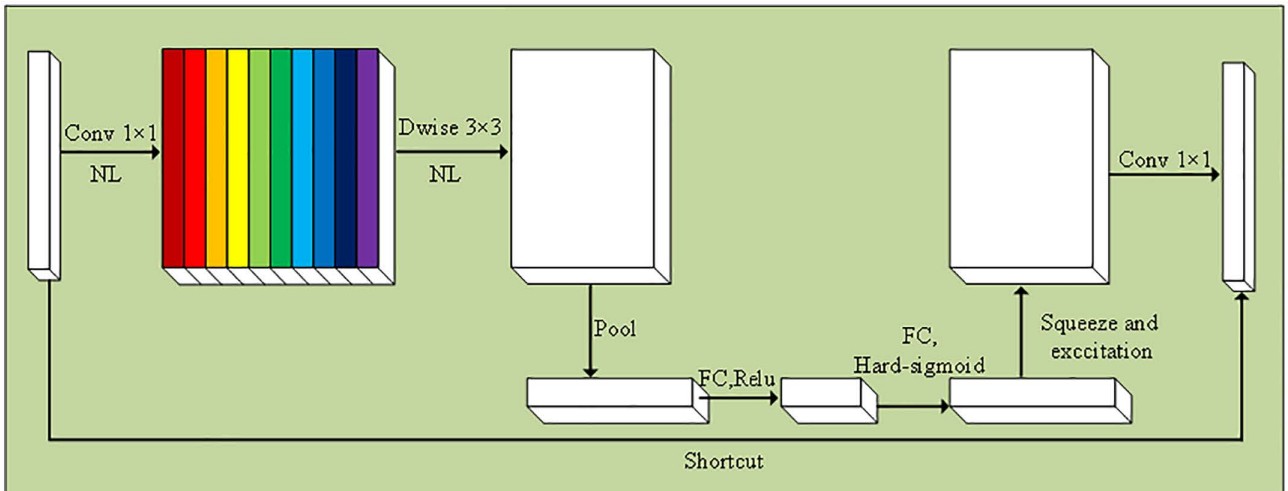

**Fig 2. Bneck module structure diagram.** Note: "Conv 1 × 1" represents the 1 × 1 convolution operation, and "NL" represents the nonlinear activation function; "Dwise 3 × 3" represents depth-separable convolution, which can significantly reduce the computational burden; "Pool", "FC, ReLU" and "FC, Hard-sigmoid" constitute the SE attention mechanism to achieve adaptive recalibration of channel weights; "Shortcut" refers to a shortcut connection, which is conducive to feature transfer and a smooth gradient. The final features are output through a 1 × 1 convolution, providing efficient expression for downstream modules.

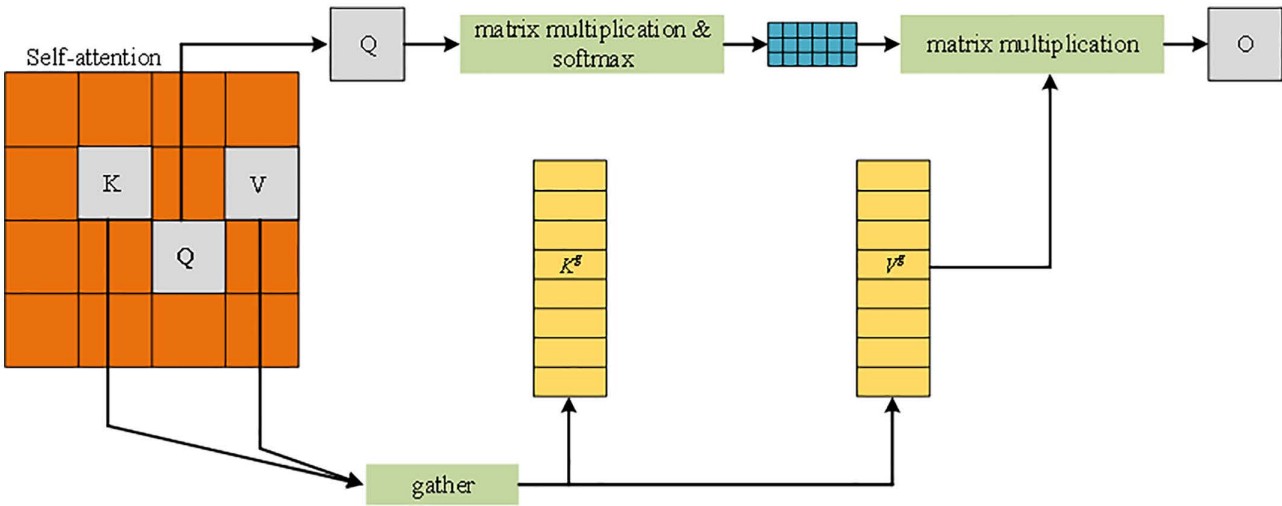

**Fig 3. BiFormer module.** Note: The input features are implemented through the Query **(Q)**, Key **(K)**, and Value (V) tensor branches to calculate the attention weights. The 'gather' module aggregates all vectors, standardizes them with Softmax, and multiplies them with Value to output the global context-enhanced features. The blue squares represent the weights of the attention matrix, the yellow vertical bars represent the sets of key/value vectors at each position, and the final output O is the self-attention result.

$$K^g = \text{gather}(K, I^r) \tag{1}$$

$$V^g = \text{gather}(V, I^r) \tag{2}$$

where $K^g$ and $V^g$ represent the collected key-value tensor, $I^r$ represents the index of the first $K$ most relevant regions in the ith row containing the ith region, and $K$ represents the collection of key-value pairs in the first $K$ relevant windows. BiFormer retains only the key and value tensor of the first $K$ most relevant regions, which effectively realizes a sparse focus on the target information and ensures a balance between detection accuracy and computational efficiency.

In the potato detection task in complex environments, by integrating the BiFormer module into the neck layer of YOLOv7-Tiny, as shown in Fig 4, the detection performance of YOLOv7-Tiny was significantly improved, mainly reflected in three aspects. the dynamic attention mechanism of BiFormer can focus on key feature areas and effectively suppress background interference; in stacked or occluded scenes, the dynamic attention mechanism significantly improves the potato detection ability of the model under complex background conditions and reduces the leakage and false detection rates; and for high-resolution images, BiFormer adaptively adjusts the attention weights of multi-scale features and can capture sharp features on small-scale targets, such as sprouting and rotting, which significantly improves detection accuracy and robustness, especially in the recognition of targets with subtle differences. In addition, the lightweight design of BiFormer avoids significant increases in computational burden, resulting in improved real-time performance and accuracy of the model.

## SimAM attention mechanisms

To further improve the detection accuracy of YOLOv7-Tiny, the simplified attention mechanism (SimAM) [33] module is introduced. SimAM calculates the autocorrelation of pixels in a feature map by local mean and variance and generates attention weights to enhance target features and suppress irrelevant background information. This involves only simple mean and element-by-element operations without additional parameters and complex matrix operations, thus improving

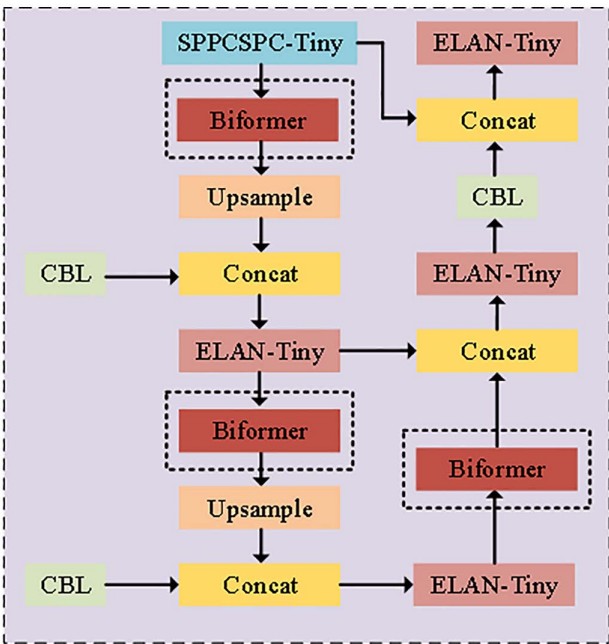

**Fig 4. Locations of BiFormer module additions.**

the model's accuracy while maintaining its lightweight properties. The structure of the SimAM attention mechanism is shown in Fig 5, where X represents the feature map, C represents the number of channels, H represents the height of the feature map, and W represents the width of the feature map.

In the YOLOv7-Tiny network structure, the SimAM module is integrated before the three common convolutional layers at the output of feature fusion, as shown in Fig 6. Through autocorrelation computation, the module strengthens the features related to sprouted and rotted potato parts, effectively reduces background noise interference, avoids complex

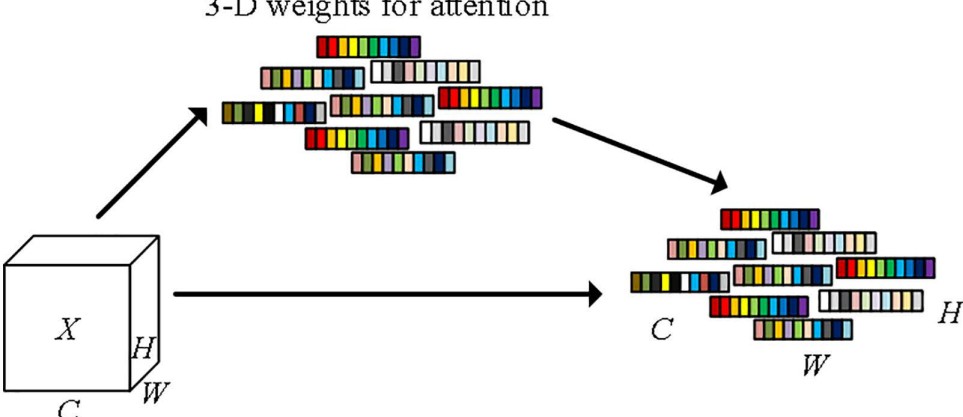

**Fig 5. Schematic diagram of the structure of the SimAM attention mechanism.** Note: The input feature tensor generates 3-D weights through the attention module. These weights are multiplied element by element with the input features to achieve feature recalibration, dynamically adjusting the response intensity of each space and channel position, thus realizing the joint attention and adaptive feature enhancement of space and channel.

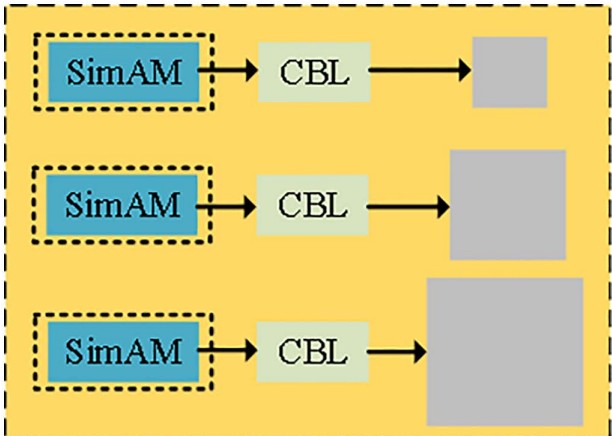

**Fig 6. SimAM Attention Mechanism add locations.**

computation and additional parameter learning, significantly improves detection without increasing model complexity, improves the robustness of the model in multi-scale information processing and overlapping target scenarios, and effectively reduces false and missed detection. The module is integrated in front of the convolutional layer after the feature fusion output, optimizing feature representation and adapting to the real-time detection requirements of high-resolution images, making it suitable for resource-constrained mobile devices or embedded systems.

Fig 7 shows the overall architecture of the proposed improved YOLOv7-Tiny object detection network. The model structure consists of three parts: the feature extraction network (backbone), the feature fusion network (neck), and the feature detection output layer (head). It further integrates modules such as MobileNetV3, BiFormer and SimAM to significantly enhance the lightweight capacity and detection performance of the model.

## The Focal-EIOU Loss Function

Focal-EIOU loss function [34] focuses on the optimization of high-quality bounding boxes by inheriting the advantages of the CIOU loss function to further improve the accuracy and convergence speed of the detection model. By combining the advantages of Focal and EIOU loss, the Focal-EIOU loss function solves the category imbalance problem and improves bounding box regression ability. Focal-EIOU loss function is defined as shown in Formula (3):

$$L_{EIOU} = L_{IOU} + L_{dis} + L_{asp} = 1 - IOU + \frac{p^2\left(b, b^{gt}\right)}{\left(w^c\right)^2 + \left(h^c\right)^2} + \frac{p^2\left(w, w^{gt}\right)}{\left(w^c\right)^2} + \frac{p^2\left(h, h^{gt}\right)}{\left(h^c\right)^2}$$

(3)

where $L_{EIOU}$ represents the loss function; $L_{IOU}$ represents the IOU loss function; $L_{dis}$ represents the distance loss; $L_{asp}$ represents the directional loss; $b, b^{gt}$ represents the center point of the prediction and target boxes; $p$ represents the Euclidean distance between the two centers; $C$ represents the diagonal distance between the prediction frame and the minimum outer frame of the actual frame; $h^c$ and $w^c$ represent the height and width of the smallest outer frame, respectively; $h$ and $w$ represent the height and width of the prediction box, respectively.

The introduction of Focal-EIOU loss function has significant advantages for improving target detection performance. By combining the advantages of EIOU and Focal loss, the Focal-EIOU loss function not only reduces the loss contribution of easy-to-classify samples by focusing on difficult-to-classify samples but also introduces a penalty term for the distance between the center of the bounding box and the aspect ratio while calculating the similarity of the aspect ratio, which

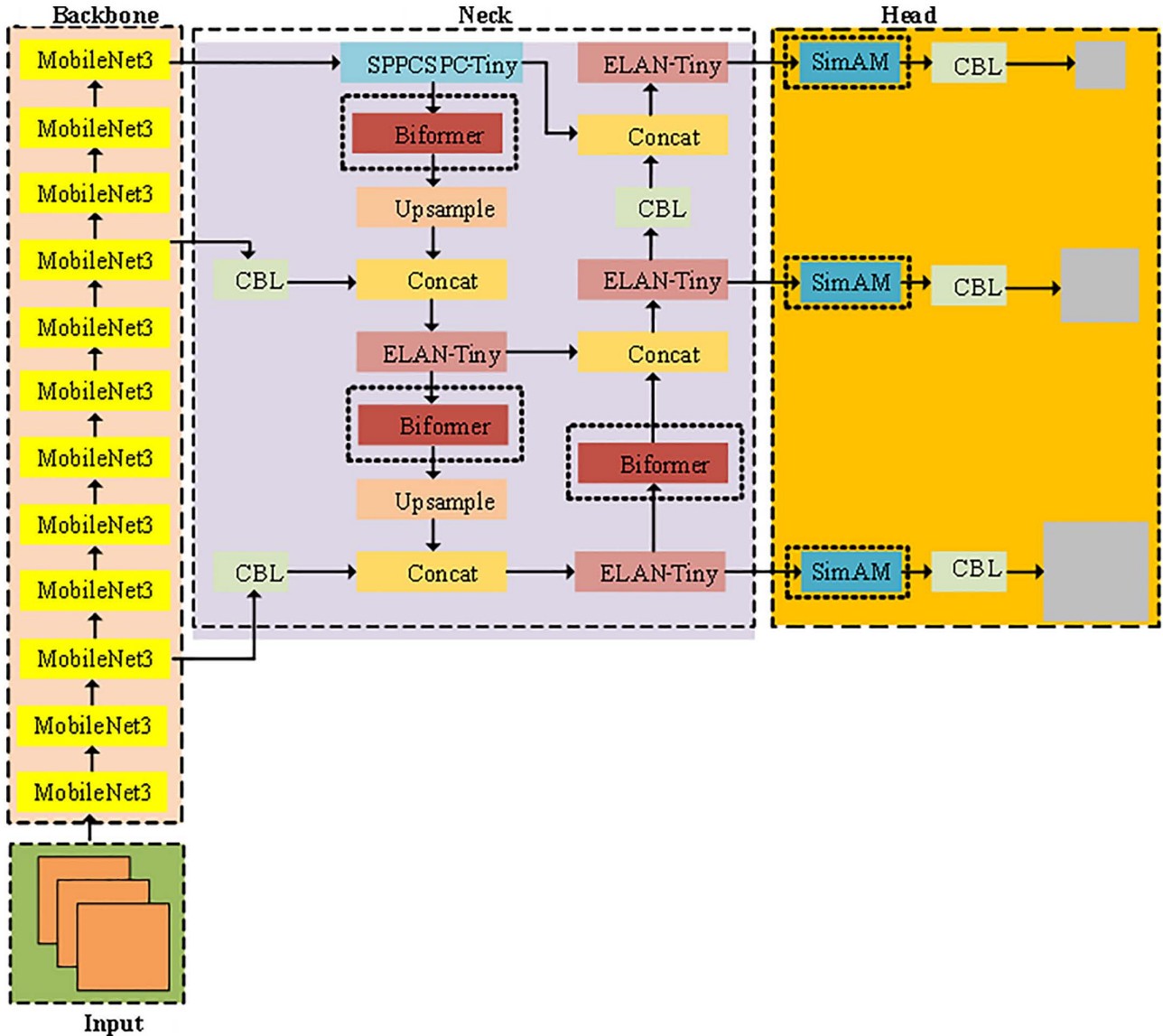

**Fig 7. Improved YOLOv7-Tiny model structure diagram.** Note: This structure is based on the YOLOv7-Tiny architecture and integrates three key improvements to enhance performance: introducing MobileNetV3 Small to replace the original Backbone, significantly reducing parameter size and computational complexity; Integrate the BiFormer attention module in the Neck to enhance the multi-scale feature fusion capability and the focus on occluded targets; Embed the SimAM spatial-channel attention mechanism before the Neck output to effectively suppress background interference and highlight key detail areas. In the figure, different color boxes represent different functional modules, and arrows indicate the direction of the feature flow. This structure ensures high detection accuracy while taking into account lightweight and real-time performance, and is suitable for embedded deployment scenarios with limited resources.

further reduces the gap between the predicted box and the real box, significantly accelerating the convergence speed of the model and improving the quality of the detected frame. Compared with the traditional loss functions, the Focal-EIOU function performs particularly well in the detection of small and occluded targets, especially in normal, sprouted, and spoiled potato high-resolution images, and can dynamically adjust loss weights to highlight abnormal samples. By reducing the interference of background noise and low-quality samples, the method improves robustness and model stability in complex environments and achieves more accurate target localization in small and occluded areas.

## Implementation details and evaluation metrics

### Introduction to the dataset

To construct a high-quality potato detection dataset, we utilized web crawler technology to obtain high-resolution images of normal, sprouted and decayed potatoes from open sources, and combined them with pre-collected samples to form a diversified dataset of 8,800 images (constructed after filtering and duplication removal). The dataset comprises images of potatoes in normal, sprouted and rotted states under a variety of scenarios, such as single categories, combination categories, adherence conditions and complex backgrounds, to ensure the robustness and generalizability of the model in various environments. Photographs of the different states are shown in Fig 8.

Owing to the diversity of image sources, resolution differences can impact detection performance. For this reason, all images were normalized to 640×640 pixels in the data preprocessing stage to improve the consistency and adaptability of the model in training and inference. The dataset was subsequently manually labeled using the LabelImg tool using three labels: "Normal", "Sprout" and "Rotten", representing normal, sprouted and rotted potatoes, respectively.

In terms of sample distribution, the proportions of the three types of images in the entire dataset are as follows: 43% normal potatoes, 29% sprouted potatoes, and 28% rotten potatoes. As images with multiple features (such as simultaneous germination and rotting) are possible, to ensure consistent data labeling and model training effectiveness, we prioritize the main features for image classification. That is, mixed feature images, based on the most prominent visual features in the image, are classified into their most representative single category by manual annotators. For example, when the

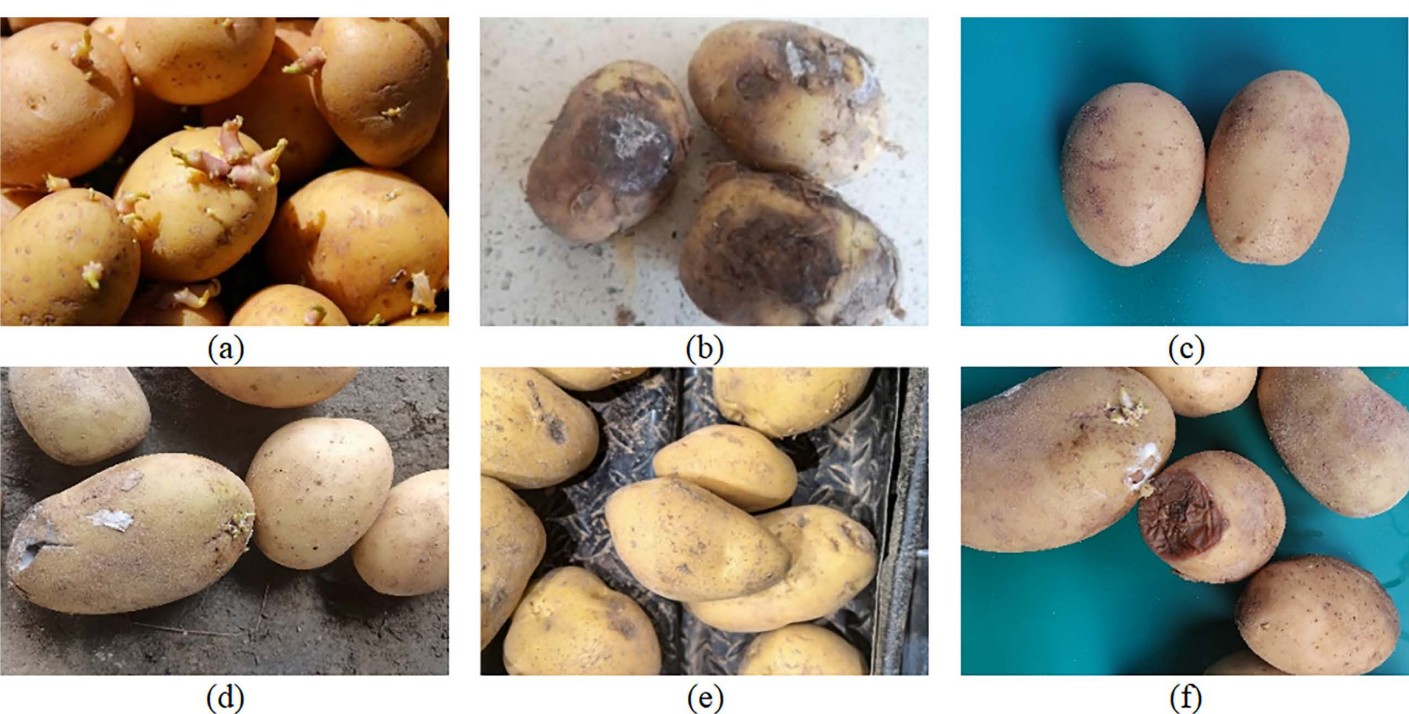

**Fig 8. Image data of potatoes in different states. (a)** Sprouting potatoes. **(b)** Spoiled potatoes. **(c)** Normal potatoes. **(d)** Adhered multigrain potatoes. **(e)** Complex environmental potatoes. **(f)** Multi-category mixed potatoes. Note: Some of the images in this study are derived from publicly accessible pages on the Internet (without login verification and access restrictions). The collection and use of image data strictly comply with the terms of the source platform and do not involve personal information, sensitive content or restricted data. The obtained image data is only used for non-commercial scientific research and model training.

rotten area dominates the visual area of an image, the image is classified into the "rotten potato" category. To ensure category balance in the model during training and robustness of the evaluation results, the data division stage adopts a stratified sampling strategy combined with 5-fold cross-validation for data organization. The overall data are divided into five subsets according to category proportion. In each round of training, four subsets are selected as the training set, and one subset is selected as the test set. Training and verification are carried out alternately in five rounds to reduce performance deviation caused by the randomness of the division. Furthermore, to enhance the training efficiency and generalization capability, in each round of the training set, the training subset and the validation subset are further divided at a ratio of 4:1 for parameter optimization and model selection, respectively, to control the risk of overfitting and ensure stable improvement of model performance.

## Experimental platform

The test environment for this test is shown in Table 1.

The hyperparameters adopted in this study include the initial learning rate, batch size, momentum coefficient, weight attenuation, etc. The initial values were set mainly with reference to the official configuration of YOLOv7 and related lightweight object detection work completed in recent years. On this basis, we tuned key hyperparameters by combining the performance of the training and validation sets through an experience-driven grid search strategy.

## Specifically, we fixed or tuned them as follows

Initial learning rate: Select from {0.01, 0.005, 0.001}, take the validation set mAP as the metric, and finally select 0.01;

Batch size: Try {4, 8, 16} within the allowable range of GPU video memory and select 8, which is stable in training and has a moderate convergence speed;

Momentum parameter: Fixed at 0.937, refer to the official configuration of YOLOv7;

Weight decay: Fixed at 0.0005;

The number of training rounds (epochs): By observing the changing trends of training losses and validation indicators, it was fixed at 200 rounds to achieve full convergence.

Notably, to reduce subjectivity in the selection of hyperparameters, we used the mean average performance (mAP) of 5-fold cross-validation as the evaluation criterion at each round of training to ensure robustness and generalizability of parameter tuning.

## Evaluation metrics

The performance of the model was evaluated using the indicators of precision, recall, mean average precision and computation. The precision rate P is the proportion of correctly predicted positive samples out of all predicted positive class samples. The recall rate R is the proportion of correctly predicted positive samples among the actual positive samples. The mean accuracy precision (mAP) is a better indicator of global performance than P and R due to their unstable fluctuations. P, R and mAP are calculated as shown in Formulas (4–6).

**Table 1. Test environment.**

| Name | Test environment |
| --- | --- |
| Deep learning framework | PyTorch 2.0.0 |
| CPU | 12th Gen Intel (R) Core (TM) i5-12400F 2.50 GHz |
| GPU | NVIDIA GE Force RTX 3060 |
| Video memory | 12 GB |
| Software version | CUDA 11.8 |

$$p = \frac{T_P}{T_P + F_P} \times 100\% \tag{4}$$

$$R = \frac{T_P}{T_P + F_N} \times 100\% \tag{5}$$

$$mAP = \int_0^1 P(R)dR \times 100\% \tag{6}$$

where $T_P$ represents the number of correct targets detected; $F_P$ represents the number of targets erroneously detected; and $F_N$ represents undetected $T_P$.

In this study, computational and parameter size are used to measure the complexity of the model, and the frame rate is used to evaluate the real-time detection performance of the model.

## Experimental results and discussion

### Ablation test

An ablation test was conducted to verify the effect of adding each module to the original YOLOv7-Tiny model to find a more suitable model for the potato sprouting and rotting recognition task. Based on the original YOLOv7-Tiny model, we modified the MobileNetV3, BiFormer and SimAM modules individually, in pairs and completely, and finally, we replaced the original loss function with Focal-EIOU loss on the fully loaded model and tested it to evaluate various indices. The results of the experiment are shown in Table 2.

According to the ablation results, the mAP of the original YOLOv7-Tiny model is 87.5%, the computational load is 13.2 G, the number of parameters is 6.0 MB, and the inference speed is 130.2 FPS. On this basis, the introduction of different modules significantly affects the detection performance in several ways: MobileNetV3 significantly reduces the number of parameters and number of calculations, reducing them to 5.4 MB and 10.9 G, respectively, and increases mAP to 88.7%; the BiFormer module increases mAP to 89.9%, with a slight reduction in the number of calculations and

Table 2. Ablation test results.

| Model | Precision/% | Recall/% | Mean Average Precision/% | Computation/GB | parameter size/MB | Frame Rate/fps |
|---|---|---|---|---|---|---|
| YOLOv7-Tiny | 86.40±0.07 | 86.10±0.08 | 87.50±0.09 | 13.20 | 6.00 | 130.20 |
| YOLOv7-Tiny+MobileNetV3 | 87.10±0.11 | 87.60±0.10 | 88.70±0.08 | 10.90 | **5.40** | 135.20 |
| YOLOv7-Tiny+BiFormer | 90.50±0.10 | 88.80±0.10 | 89.90±0.09 | 11.30 | 5.90 | 134.40 |
| YOLOv7-Tiny+SimAM | 90.70±0.07 | 89.80±0.8 | 89.60±0.07 | 12.30 | 6.00 | 138.60 |
| YOLOv7-Tiny+MobileNetV3＋BiFormer | 90.80±0.10 | 88.40±0.12 | 88.90±0.13 | **9.80** | 5.70 | 134.30 |
| YOLOv7-Tiny+MobileNetV3＋SimAM | 91.40±0.12 | 90.30±0.11 | 89.70±0.13 | 10.50 | 5.80 | 135.30 |
| YOLOv7-Tiny+BiFormer+SimAM | 92.00±0.11 | 91.20±0.09 | 90.10±0.10 | 12.10 | 5.90 | 143.20 |
| YOLOv7-Tiny+MobileNetV3＋BiFormer+SimAM | 91.60±0.12 | 90.90±0.11 | 89.90±0.09 | 10.10 | 5.80 | 140.70 |
| YOLOv7-Tiny+MobileNetV3＋BiFormer+SimAM +Focal-EIOU Loss (LPD-YOLOv7-Tiny) | **92.30±0.08** | **91.70±0.08** | **90.30±0.07** | 10.10 | 5.80 | **142.50** |

Note: All performance metrics reported in this article, including detection precision (precision), recall (recall), and mean average precision (mAP), are derived from evaluation results on the test set, not the training or validation set.

parameters. The SimAM module improves feature resolution and increases mAP to 89.60%. The combination of Mobile-NetV3 + BiFormer achieves a balance between being lightweight and having good precision, with mAP reaching 88.9%. MobileNetV3 + SimAM significantly improves accuracy, with mAP reaching 89.7% and has a relatively low computational load, further balancing resource consumption and detection. BiFormer + SimAM improves accuracy and increases mAP to 90.1%. The combination of MobileNetV3 + BiFormer + SimAM achieves a good balance between being lightweight and having high precision, with mAP increasing to 89.9%. Based on this combination, the Focal-EIOU loss function was added to fully optimize the YOLOv7-Tiny model. Finally, the improved model (LPD-YOLOv7-Tiny) maintained a smaller parameter volume (5.8 MB) and a lower computational cost (10.1 G), relatively high detection accuracy (mAP increased from 87.5% to 90.3%) and faster inference speed (142.5 FPS), and its overall performance is excellent.

## Comparative tests

The performance of the LPD-YOLOv7-Tiny model was compared with that of mainstream models such as the basic YOLO series, SSD and Faster-RCNN in multiple dimensions. The test results are shown in Table 3 and Fig 9. YOLOv5s and YOLOv7s achieve a good balance in terms of accuracy and recall rate, reaching mAP values of 87.20% and 88.8%, respectively, but have a relatively high number of calculations and parameters. SSD and Faster R-CNN not only have high computational complexity and a large number of parameters but also relatively low accuracy and slow inference speed, which limits their application in tasks with high real-time requirements. YOLOv7-Tiny and YOLOv8-Nano have certain advantages in terms of computational complexity and number of parameters, but their detection accuracies are relatively low. YOLOv8s and YOLOv9 achieve a relatively high inference speed while maintaining high accuracy, but the number of calculations and parameters are greater than those of YOLOv7-Tiny. In contrast, the detection performance of the LPD-YOLOv7-Tiny model is significantly improved: mAP is increased to 90.3%, the inference speed reaches 142.5 fps, and the number of calculations and parameters decrease to 10.10 GB and 5.8 MB, respectively. Compared with the other basic models tested, LPD-YOLOv7-Tiny has better performance and broader application potential.

## Test effects

Compared with the basic YOLOv7-Tiny model, the LPD-YOLOv7-Tiny model achieved greater accuracy in potato defect detection. The comparative test effects are shown in Fig 10. The figure shows the generally higher label frame confidence of the LPD-YOLOv7-Tiny model, especially in the identification of the "Sprout" and "Rotten" areas of potatoes. In addition, the LPD-YOLOv7-Tiny model better identifies the characteristics of different categories of potatoes, is more sensitive in the detection of "Rotten" areas, and can identify rotten areas in a more stable manner. The LPD-YOLOv7-Tiny model,

Table 3. Comparative test results.

| Model | Precision/% | Recall/% | Mean Average Precision/% | Computation/GB | parameter size/MB | Frame Rate/fps |
|---|---|---|---|---|---|---|
| YOLOv5s | 89.60 ± 0.05 | 89.60 ± 0.06 | 87.20 ± 0.05 | 15.90 | 6.80 | 70.20 |
| YOLOv7-Tiny | 86.40 ± 0.07 | 86.10 ± 0.08 | 87.50 ± 0.09 | 13.20 | 6.00 | 130.20 |
| SSD | 87.20 ± 0.07 | 85.30 ± 0.09 | 86.30 ± 0.10 | 105.10 | 54.30 | 85.30 |
| Faster-RCNN | 79.60 ± 0.08 | 78.30 ± 0.07 | 82.40 ± 0.06 | 365.20 | 167.10 | 20.40 |
| YOLOv7s | 91.70 ± 0.07 | 89.50 ± 0.06 | 88.80 ± 0.10 | 104.10 | 61.30 | 125.30 |
| YOLOv8-Nano | 84.30 ± 0.12 | 83.10 ± 0.10 | 84.80 ± 0.10 | 12.90 | 6.10 | 134.30 |
| YOLOv8s | 89.75 ± 0.07 | 88.60 ± 0.08 | 89.30 ± 0.09 | 28.60 | 16.40 | **183.60** |
| YOLOv9 | 89.85 ± 0.08 | 89.70 ± 0.08 | 89.40 ± 0.07 | 26.30 | 13.40 | 100.60 |
| YOLOv7-Tiny+MobileNetV3 + BiFormer+SimAM +Focal-EIOU Loss (LPD-YOLOv7-Tiny) | **92.30 ± 0.08** | **91.70 ± 0.08** | **90.30 ± 0.07** | **10.10** | **5.80** | 142.50 |

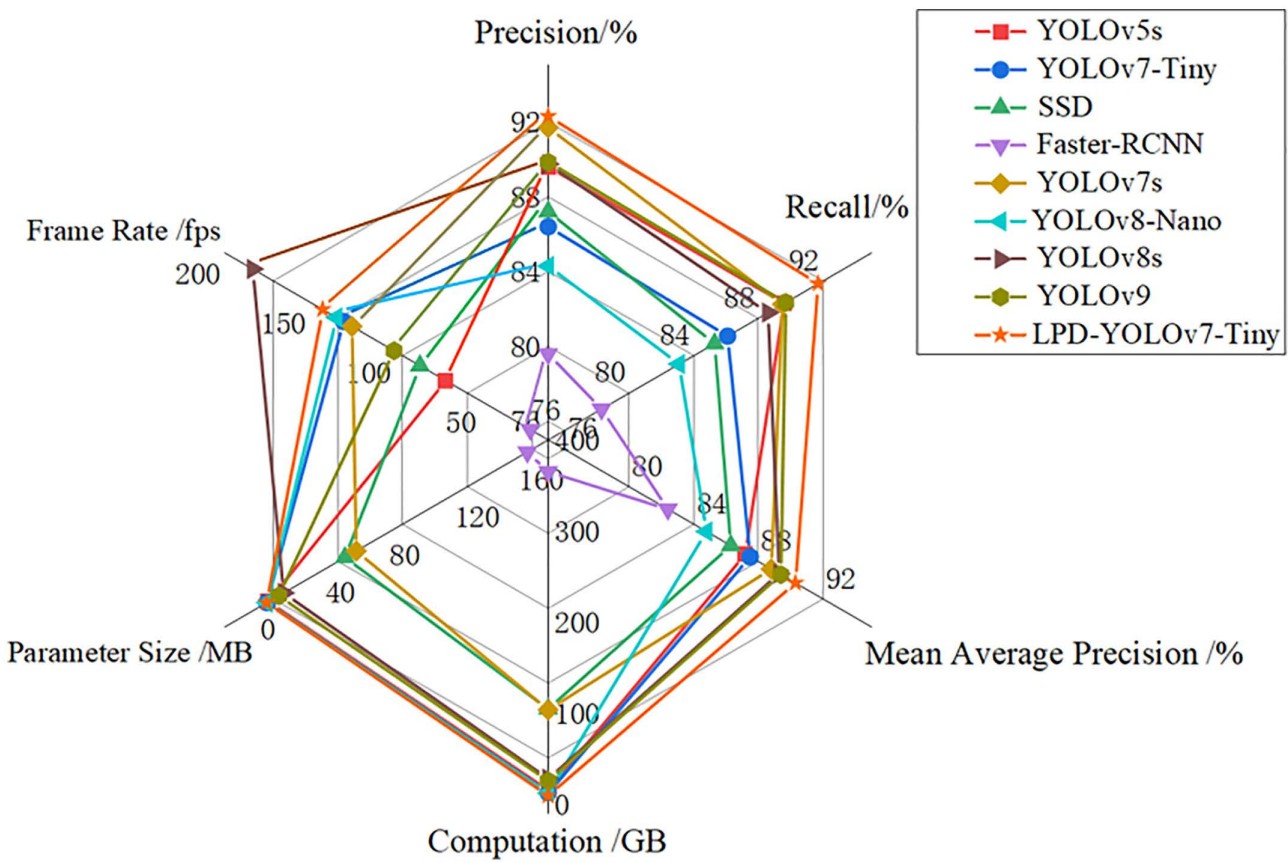

**Fig 9. Radar chart of comparative testing.**

through enhanced feature extraction and loss function optimization, has better robustness in fuzzy or complex detection scenarios.

Comparing the recognition performance of the improved model for potatoes in three states, as shown in Table 4, the LPD-YOLOv7-Tiny model has the best detection performance for normal potatoes, with precision, recall and AP reaching 92.4%, 91.6% and 90.8%, respectively. LPD-YOLOv7-Tiny's recognition accuracy on sprouted and rotten potatoes is also at a relatively high level. The average AP of the three types is 90.3%, which is consistent with the overall mAP. These findings indicate that the proposed model has good category adaptability and generalizability for complex and variable detection tasks.

In general, the LPD-YOLOv7-Tiny model, formed by introducing MobileNetV3, BiFPN, SimAM and Focal-EIOU loss optimization into the YOLOv7-Tiny model, enhances the detection accuracy and stability of different types of potato defect characteristics and effectively reduces instances of missed and false detection. Thus, LPD-YOLOv7-Tiny shows higher reliability than YOLOv7-Tiny.

## Discussion

The LPD-YOLOv7-tiny model for potato defect detection proposed in this paper effectively improves detection accuracy and positioning capability, significantly reduces computational complexity and the number of parameters, and makes the model more suitable for scenarios with limited resources and real-time detection requirements. Compared with traditional

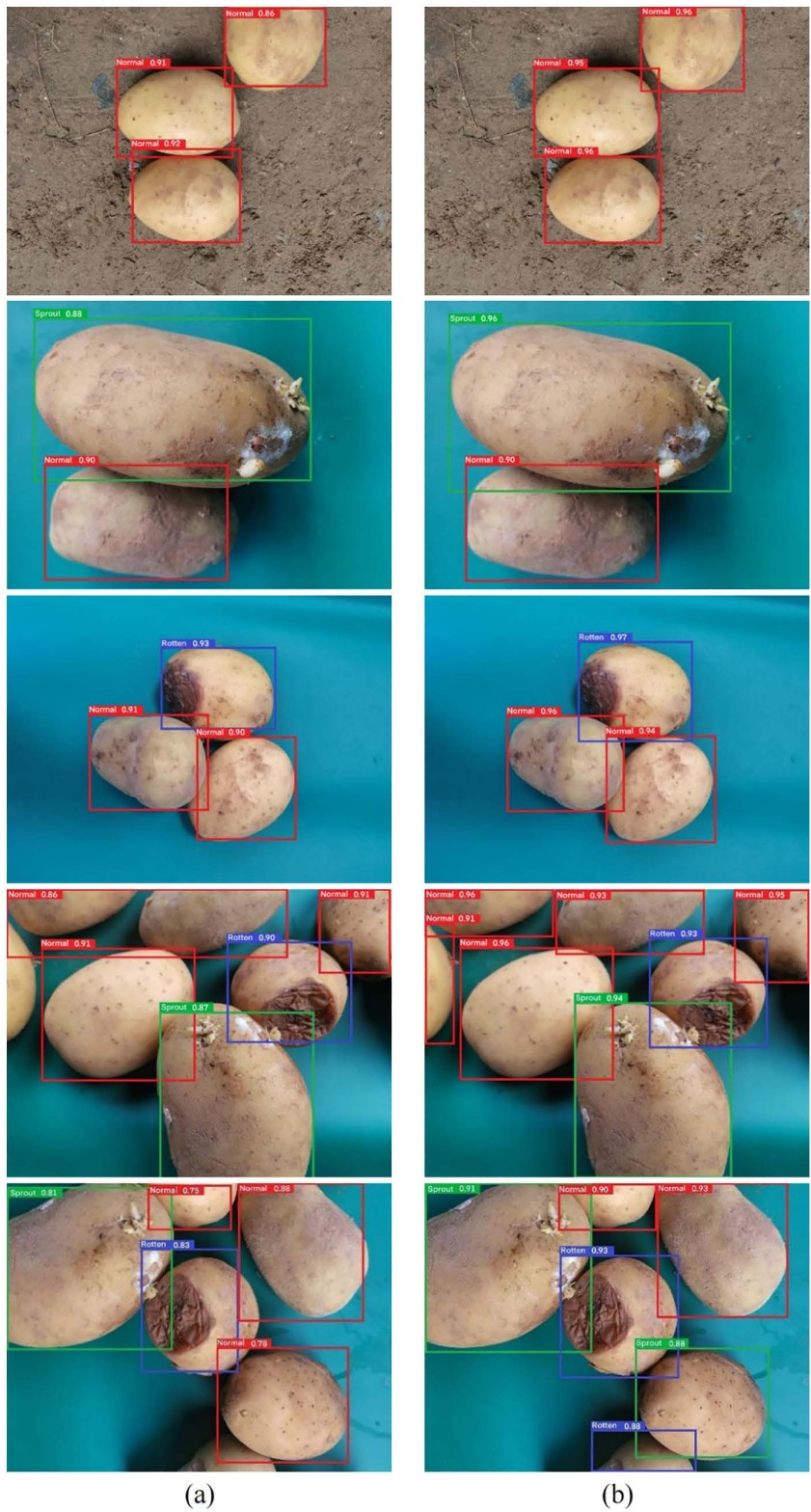

**Fig 10. Potato image recognition effects in three states. (a)** YOLOv7-Tiny. **(b)** LPD-YOLOv7-Tiny.

**Table 4. Recognition performance of the LPD-YOLOv7-tiny model for potatoes in different states.**

| State | Precision/% | Recall/% | Average Precision/% |
|---|---|---|---|
| Normal | 92.4±0.07 | 91.6±0.04 | 90.8±0.06 |
| Sprout | 90.7±0.02 | 90.1±0.04 | 90.3±0.01 |
| Rotten | 89.7±0.05 | 89.0±0.03 | 89.8±0.02 |

detection models such as SSD and Faster R-CNN, the optimization model has obvious advantages in terms of comprehensive performance and is particularly suitable for deployment on embedded devices or mobile platforms. However, this study also reveals several challenges worthy of further exploration.

First, although the combination of modules such as MobileNetV3, BiFormer and SimAM significantly enhanced the overall performance of the model, during the actual testing process, we found that the model still had problems such as decreased accuracy or even missed detection under complex background conditions. For example, when many potatoes were stacked, the model tended to merge adjacent targets into one detection box or failed to identify partially severely occluded individuals. As shown in Fig 11. The emergence of these failure cases suggests that we need to study more refined multiscale feature fusion methods or design more effective attention mechanisms to enhance the robustness of the model in fine-grained object discrimination and high-occlusion scenarios.

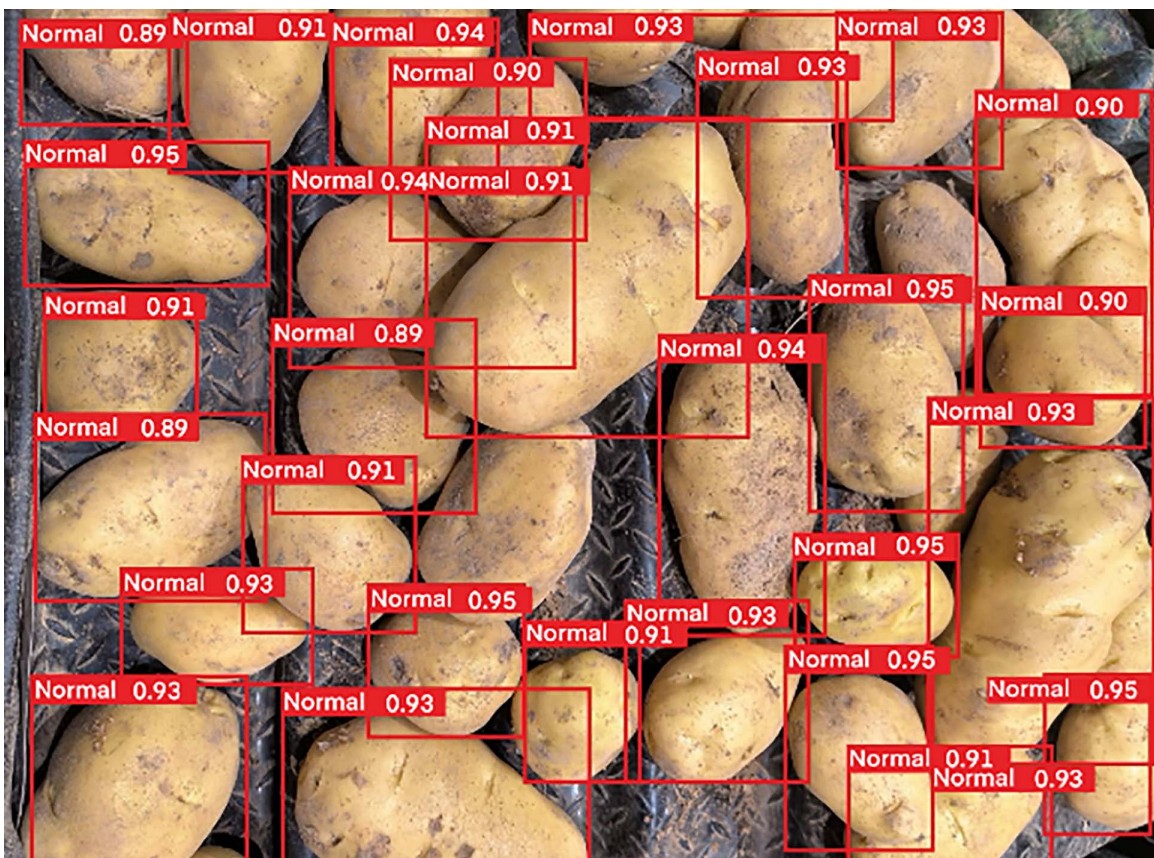

**Fig 11. Stacked potato detection.**

Second, the focal-EIOU loss module demonstrated better performance in optimizing bounding box regression, especially with a significant improvement in the localization effect for smaller targets. However, focal-EIOU loss may lead to slower model convergence speed under certain conditions, especially on datasets with low target density or an unbalanced target scale distribution. In the future, it is possible to further explore how to introduce dynamic loss weight allocation or multitask joint learning strategies to alleviate the deficiency of having a single loss function in the universality of multiple scenarios, thereby further balancing model training efficiency and detection performance.

Third, although the combination of modules achieved good performance improvements, some unexpected module interactions also emerged. For example, the BiFormer module may produce certain feature redundancy or overfitting phenomena in some low-contrast image backgrounds, thereby weakening the generalization ability of the model to a wider range of scenes. In future research, we need to explore the interaction mechanism of different modules in the feature space in greater depth and optimize the structural design of module combinations.

Then, although we have demonstrated through the compression of model computational load and parameter scale that the model is suitable for resource-constrained scenarios, there is still a lack of systematic evaluation of deployment characteristics (such as latency and energy consumption). In response to this limitation, subsequent research will focus on testing the inference speed and power consumption level of the model on typical edge devices such as Jetson Nano, and conduct real-time performance benchmarking and energy consumption simulation analysis in combination with typical load conditions. We will also explore deployment optimization strategies such as pruning, quantification, and adaptive reasoning mechanisms to further enhance the efficiency and practicality of the model in real-world application environments.

Furthermore, although this study constructed diverse datasets and adopted 5-fold cross-validation to enhance model robustness, the generalization performance has not yet been evaluated on completely external datasets. Due to the lack of a publicly available standardized dataset of potato quality images and the differences in varieties, shooting conditions and labeling methods among data from different sources, external verification cannot be completed at the current stage. Future work will consider collecting or introducing heterogeneous data to further verify the model's adaptability and scalability under cross-environmental and cross-device conditions.

## Conclusion

The improved YOLOv7-Tiny model for real-time potato quality detection proposed in this study significantly improves the accuracy and real-time performance of lightweight target detection by integrating MobileNetV3 small, BiFormer, SimAM modules and the Focal-EIOU loss function. Thus, it provides an efficient solution for detection tasks in high-precision and low-resource environments.

(1) By introducing MobileNetV3 small, the parameter count and computational complexity of the model were significantly reduced. The BiFormer module effectively enhanced multi-scale feature fusion and target focusing capabilities in complex background scenarios, whereas SimAM highlighted important local features and suppressed redundant information. In addition, the Focal-EIOU loss function further improved the accuracy of bounding box positioning of the model.

(2) The proposed improved model demonstrated outstanding performance in the detection of normal, sprouted and rotten potatoes, increasing mAP to 90.3%. Moreover, the number of model parameters was reduced to 5.8 MB, and the computational cost was decreased to 10.1 G, successfully achieving an ideal balance of high precision, low complexity and fast reasoning (142.5 fps).

(3) Compared with mainstream models such as the YOLO series, SSD and Faster-RCNN, the model optimized in this study has obvious advantages in terms of accuracy and inference speed, showing particularly broad application potential in resource-constrained environments.

In the future, more advanced feature fusion and attention mechanisms can be explored to further enhance the robustness of the model in occlusion and target stacking scenarios. Research is also being conducted on the deployment efficiency and energy consumption of the model to extend it to more application fields, such as medical image analysis, multi-variety detection in agriculture, and environmental perception in autonomous driving.

## Author contributions

**Conceptualization:** Hong Yu.

**Data curation:** Hong Yu, Jiaxuan Hao.

**Funding acquisition:** Yongbo Li.

**Methodology:** Hong Yu.

**Project administration:** Hong Yu, Yongbo Li.

**Software:** Yongbo Li.

**Validation:** Hong Yu.

**Writing – original draft:** Hong Yu.

**Writing – review & editing:** Jiaxuan Hao, Yongbo Li.

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
