## [Decision Letter · Decision Letter 0]

25 Apr 2025

Dear Dr. Li,

We look forward to receiving your revised manuscript.

Kind regards,

Fatih Uysal, Ph.D.

Academic Editor

PLOS ONE

Journal Requirements:

2. In your Methods section, please include additional information about your dataset and ensure that you have included a statement specifying whether the collection and analysis method complied with the terms and conditions for the source of the data.

“The Basic science (Natural science) research project in universities of Jiangsu (24KJB510045); The 2022 Taizhou “Fengcheng Talent Program” Young science and technology talent Lifting Project (Taizhou Association for Science and Technology Document (2022) No. 64); The Jiangsu Agri-animal Husbandry Vocational College Research Project (NSF2023ZR12).”

5. We note that your Data Availability Statement is currently as follows: All relevant data are within the manuscript and its Supporting Information files.

6. PLOS requires an ORCID iD for the corresponding author in Editorial Manager on papers submitted after December 6th, 2016. Please ensure that you have an ORCID iD and that it is validated in Editorial Manager. To do this, go to ‘Update my Information’ (in the upper left-hand corner of the main menu), and click on the Fetch/Validate link next to the ORCID field. This will take you to the ORCID site and allow you to create a new iD or authenticate a pre-existing iD in Editorial Manager.

**Additional Editor Comments:**

Please revise the paper based on reviewer comments.

Reviewers' comments:

Reviewer's Responses to Questions

**Comments to the Author**

1. Is the manuscript technically sound, and do the data support the conclusions?

Reviewer #1: Yes

Reviewer #2: Partly

Reviewer #3: Yes

2. Has the statistical analysis been performed appropriately and rigorously?

Reviewer #1: Yes

Reviewer #2: No

Reviewer #3: Yes

3. Have the authors made all data underlying the findings in their manuscript fully available?

Reviewer #1: No

Reviewer #2: No

Reviewer #3: No

4. Is the manuscript presented in an intelligible fashion and written in standard English?

Reviewer #1: Yes

Reviewer #2: No

Reviewer #3: Yes

Reviewer #1: 1. Include comparison with other recent lightweight detectors such as YOLOv8-Nano or MobileSAM for a better positioning of your model strengths.

2. Add k-fold cross-validation or statistical significance tests (e.g., confidence intervals) to support the robustness of your mAP improvement claims.

3. The Focal-EIOU addition is said to improve accuracy, but it may slow convergence. Provide training time comparisons and mention any increased training cost.

4. Include a combined flowchart summarizing how MobileNetV3, BiFormer, SimAM, and Focal-EIOU fit together in the architecture (in addition to the YOLOv7-tiny base diagram).

5. Suggesion change title to: LPD-YOLOv7-Tiny: An Enhanced Lightweight YOLOv7-Tiny Model for Real-Time Potato Quality Detection

Reviewer #2: introduction

The introduction clearly highlights the motivation for improving potato quality detection but is wordy and lacks focus in some areas. Many sentences are overly long, and several citations are clustered without sufficient explanation (e.g., references [4–15]). The literature review mixes older and newer methods without organizing them thematically (e.g., traditional vs deep learning-based).

Methods

The methodological section is rich in technical detail, which is commendable, but it would benefit greatly from better structure and clarity. Each sub-module (MobileNetV3-small, BiFormer, SimAM, Focal-EIOU) is described in isolation without a summary table or figure comparing their benefits and trade-offs. The mathematical formulations are helpful but lack context—for instance, defining variables without explaining their physical interpretation can confuse readers unfamiliar with the concepts. It is recommended to add a schematic of the full LPD-YOLOv7-Tiny architecture and clearly mark the points of module integration. A summary diagram showing the impact of each module on speed, accuracy, and model size would greatly enhance understanding.

Experiments & Results

The experimental design is mostly sound, and the ablation and comparison tables are informative. However, the lack of statistical rigor significantly weakens the validity of the conclusions. The results are presented as single-run metrics without any mention of multiple trials, standard deviation, or confidence intervals. To address this, the authors should rerun key experiments (especially ablation studies) at least three times and report mean ± standard deviation. Additionally, the authors should conduct significance testing (e.g., t-tests or Wilcoxon signed-rank tests) to confirm that performance gains are not due to random variation. Adding plots (e.g., bar charts with error bars) would also help visualize improvements across models.

Figures and Tables

The figures included are functionally useful but lack quality and clarity. For example, the model architecture diagrams are low-resolution and difficult to interpret, especially in print. Figure legends are minimal and do not explain the key takeaway from each figure. To enhance the visual presentation, ensure all figures are vector-based and readable at publication scale. Each figure should have a fully descriptive caption. For tables, particularly Table 2 and 3, bold or highlight the best-performing models to guide the reader’s attention. Consider adding a visual summary or radar chart comparing your model to others in terms of accuracy, speed, and parameter size.

Discussion

The discussion reiterates results but offers limited critical reflection. It lacks an honest assessment of the model's limitations and opportunities for further enhancement. For example, while detection in complex backgrounds is mentioned as a challenge, no concrete failure cases are described. The authors should discuss where the model underperforms (e.g., severe occlusion, overlapping potatoes), how module combinations might interact in unanticipated ways, and how the model could be adapted for other agricultural products. Additionally, implications for deployment (e.g., latency on edge devices, energy consumption) could be elaborated upon. A dedicated subsection on “Limitations and Future Work” would add value.

Conclusion

The conclusion effectively summarizes the model's improvements but is too repetitive and lacks a strong forward-looking statement. It reiterates the same metrics listed earlier without synthesizing the overall impact of the work. To improve, distill the main finding into one powerful sentence, followed by key takeaways

Reviewer #3: Dear Author(s):

This study proposes a real-time, lightweight, and accurate detection model (LPD-YOLOv7-Tiny) for classifying potatoes as normal, sprouted, or rotten by enhancing the YOLOv7-Tiny architecture. The integration of MobileNetV3, BiFormer, and SimAM modules aims to improve detection accuracy and model efficiency. Furthermore, the Focal-EIOU loss function enhances bounding box quality. While the method is valuable, addressing the following issues would significantly strengthen the study.

1. The literature should be expanded. Related works utilizing YOLO architectures for similar detection problems are missing.

2. The aim of the study was not to develop a device for the detection of a real-time problem. Why YOLOv7-tiny was preferred. Isn't it more logical to make architectural improvements on the YOLOv7 architecture by prioritizing performance according to the Bag of Specials (BoS) logic?

3. The change in parameter count should be explicitly stated. Comparisons before and after optimization are missing.

4. The class distribution in the dataset ("Normal", "Sprout" and "Rotten") should be presented, and whether stratified sampling was applied should be clarified.

5. The hyperparameter selection method must be specified. Was it manual, based on prior work, or via algorithmic tuning?

6. How were the hyperparameters tuned for SSD and Faster-RCNN? This is crucial for fair comparison.

7. The architecture diagram of the proposed model is missing. Integration of each module should be illustrated.

8. The proposed model architecture (LPD-YOLOv7-Tiny) is not given schematically. The integration of the blocks should be clearly shown.

9. Table 3 lacks comparison with recent YOLO versions (SOTA-v9–v12).

10. Are the reported Precision, Recall, and mAP values from the test set? If from training, this must be stated.

11. Class-wise metrics are not provided. Metrics for “Normal,” “Sprout,” and “Rotten” should be separately reported.

12. No internal validation or external validation was performed in this study. At least “cross-validation” in the name of internal validation and measuring the generalization ability of the proposed model would add strength to the study.

13. Visual comparisons are limited. More examples, especially for sprout/rotten detection, should be shown.

The work overall presents a meaningful contribution. However, addressing the above shortcomings will significantly enhance its impact both in applied and academic contexts.

**Do you want your identity to be public for this peer review?** For information about this choice, including consent withdrawal, please see our Privacy Policy

Reviewer #1: No

Reviewer #2: **Yes: ** aasim ayaz wani

Reviewer #3: No

---

## [Author Response · Author response to Decision Letter 1]

7 Jul 2025

We provided detailed responses to the questions raised by the reviewers and editors. The contents of the responses were all uploaded in the form of documents to the attached file for the experts to review.

---

## [Decision Letter · Decision Letter 1]

18 Jul 2025

Dear Dr. Li,

We look forward to receiving your revised manuscript.

Kind regards,

Fatih Uysal, Ph.D.

Academic Editor

PLOS ONE

Journal Requirements:

Additional Editor Comments:

Kindly revise your manuscript based on the reviewers' feedback.

Reviewers' comments:

Reviewer's Responses to Questions

**Comments to the Author**

Reviewer #1: (No Response)

Reviewer #2: All comments have been addressed

Reviewer #3: All comments have been addressed

2. Is the manuscript technically sound, and do the data support the conclusions?

Reviewer #1: (No Response)

Reviewer #2: Yes

Reviewer #3: Yes

3. Has the statistical analysis been performed appropriately and rigorously?

Reviewer #1: (No Response)

Reviewer #2: Yes

Reviewer #3: Yes

4. Have the authors made all data underlying the findings in their manuscript fully available?

Reviewer #1: (No Response)

Reviewer #2: Yes

Reviewer #3: Yes

5. Is the manuscript presented in an intelligible fashion and written in standard English?

Reviewer #1: (No Response)

Reviewer #2: Yes

Reviewer #3: Yes

Reviewer #1: Expand on failure cases with images and analysis

Briefly comment on real-world deployment feasibility

Reviewer #2: The authors have approached a niche yet important problem with rigor, technical depth, and a strong sense of practical relevance. The integration of MobileNetV3-small, BiFormer, SimAM, and Focal-EIOU into the YOLOv7-Tiny framework demonstrates a solid understanding of the challenges of deploying models in constrained environments. The authors demonstrate commendable rigor in model design and evaluation, and have been responsive to previous reviewer comments, incorporating comparative baselines (YOLOv8, YOLOv9), statistical testing, and clearer visualizations.

However, there are some improvements that i would suggest :-

1. Writing Clarity and Conciseness

While technically accurate, the manuscript tends to be wordy and sometimes repetitive (especially in Sections 1 and 5). Consider tightening the prose and limiting restatements of performance metrics unless adding new context.

2. External Validation Missing

The study would be further strengthened by evaluation on a completely external dataset (e.g., from a different environment, device, or potato variety). Even a small external test set would help assess generalizability. If that is not possible, please address this in the limitations/future directions section.

3. Model Failures and Interpretability

The discussion notes challenges under occlusion and dense stacking, but no visual examples are provided. Including 1–2 side-by-side examples of successful vs. failed detection would enhance reader understanding of limitations.

4. Deployment Details

While edge deployment is discussed conceptually, no practical profiling is included. It would strengthen the manuscript to briefly suggest how this approach could be validated—such as testing the model on devices like the Jetson Nano or simulating inference time and energy consumption—to better align with the manuscript’s real-time deployment focus.

5. Figure Captions and Layout

Some model diagrams remain dense despite improvements. Breaking them into smaller subfigures or adding callouts could improve comprehension. Captions should also summarize the interpretive takeaway rather than just describing components.

This paper presents a meaningful and well-validated contribution to lightweight agricultural vision systems. With minor polishing of writing and figures—and, ideally, inclusion of additional failure case visualizations or external validation—it will be a valuable addition to the field.

Reviewer #3: The additional details you provided on your hyper-parameter settings make it clear that the comparison is fair, especially your use of grid-search based on 5-fold cross-validation on the same hardware/resolution/epoch conditions is convincing. Your choice of LR = 0.005 for SSD and LR = 0.01 for Faster-RCNN, as well as your standardization of batch size = 8 and NMS = 0.5, reinforces the methodological consistency of your results. This clarity increases the value of the paper as it makes it easier for readers to replicate; Good luck with your work

**Do you want your identity to be public for this peer review?** For information about this choice, including consent withdrawal, please see our Privacy Policy

Reviewer #1: No

Reviewer #2: **Yes: ** Aasim Ayaz Wani

Reviewer #3: No

---

## [Author Response · Author response to Decision Letter 2]

24 Aug 2025

All the reviewers' questions are replied in the form of attachments. For details, please refer to the attachments.

---

## [Editor Report · Decision Letter 2]

26 Aug 2025

LPD-YOLOv7-Tiny: An Enhanced Lightweight YOLOv7-Tiny Model for Real-Time Potato Quality Detection

PONE-D-25-10841R2

Dear Dr. Li,

We’re pleased to inform you that your manuscript has been judged scientifically suitable for publication and will be formally accepted for publication once it meets all outstanding technical requirements.

Kind regards,

Fatih Uysal, Ph.D.

Academic Editor

PLOS ONE

Additional Editor Comments (optional):

The decision to accept the article was made on the grounds that the revisions sufficiently addressed the referee comments and that the article demonstrates significant potential to contribute to the literature.
---

## [Editor Report · Acceptance letter]

PONE-D-25-10841R2

PLOS ONE

Dear Dr. Li,

I'm pleased to inform you that your manuscript has been deemed suitable for publication in PLOS ONE. Congratulations! Your manuscript is now being handed over to our production team.

Kind regards,

on behalf of

Dr. Fatih Uysal

Academic Editor

PLOS ONE